



# How Does Cloud-Radiative Heating over the North Atlantic Change with Grid Spacing, Convective Parameterization, and Microphysics Scheme?

Sylvia Sullivan[1,2], Behrooz Keshtgar[2], Nicole Albern[2,3], Elzina Bala[2], Christoph Braun[2], Anubhav Choudhary[2], Johannes Hörner[2,4], Hilke Lentink[2], Georgios Papavasileiou[2,5], and Aiko Voigt[2,4]

[1]Department of Chemical and Environmental Engineering, University of Arizona, Tucson, Arizona
[2]Insitute of Meteorology and Climate Research, Karlsruhe Institute of Technology, Karlsruhe, Germany
[3]Aon Versicherungsmakler Deutschland GmbH, Hamburg, Germany
[4]Department of Meteorology and Geophysics, University of Vienna, Vienna, Austria
[5]National Observatory of Athens, Institute for Environmental Research and Sustainable Development, Lofos Koufou, Penteli, Greece

**Correspondence:** S. Sullivan (sylvia@arizona.edu) and A. Voigt (aiko.voigt@univie.ac.at)

**Abstract.** Cloud-radiative heating (CRH) within the atmosphere and its changes with warming affect the large-scale atmospheric wind patterns in a myriad of ways, such that reliable predictions and projections of circulation require reliable calculations of CRH. In order to assess sensitivities of upper-tropospheric midlatitude CRH to model settings, we perform a series of simulations with the Icosahedral Nonhydrostatic Model (ICON) over the North Atlantic using six different grid spacings, parameterized and explicit convection, and one- versus two-moment cloud microphysics. While sensitivity to grid spacing is limited, CRH profiles change dramatically with microphysics and convection schemes. These dependencies are interpreted via decomposition into cloud classes and examination of cloud properties and cloud-controlling factors within these different classes. We trace the model dependencies back to differences in the mass mixing ratios and number concentrations of cloud ice and snow, as well as vertical velocities. Which frozen species are radiatively active and the coupling of microphysics and convection schemes turn out to be crucial factors in altering the modeled CRH profiles.

## 1 Introduction

Clouds have important radiative effects within the atmosphere. They absorb the outgoing infrared radiation that would otherwise escape to space and reemit it at colder temperatures. They also absorb and reflect incoming solar radiation that would otherwise warm the atmosphere and surface. The relative balance of these warming and cooling effects depends on the cloud phase and altitude. The cooling effect tends to dominate for low-level liquid clouds, whereas the warming effect tends to dominate for high-level ice clouds.

Within the atmosphere, the impact of clouds on atmospheric radiation is generally quantified with *cloud-radiative heating* rates, as this heating is what influences circulation. This cloud-radiative heating can be calculated as the difference between all-sky and clear-sky flux divergences. A local heating or cooling rate due to clouds translates to changes in atmospheric



temperature and pressure gradients and, hence, driving force for winds. The notion that clouds are not only embedded in the circulation but also determine it has become an important theme in recent years within clouds and climate research (e.g., Bony et al., 2015; Voigt and Shaw, 2015; Voigt et al., 2020).

A burgeoning body of work highlights the many ways in which clouds affect circulation via their radiative heating. Warming in the upper troposphere due to ice clouds can increase tropical stability, driving poleward expansion of the large-scale
circulation (Lu et al., 2007). In the tropics, cloud-radiation interactions cause tightening of the ascent region and expansion of the descent region within the Hadley cell (Albern et al., 2018). Radiative heating from tropical upper-tropospheric clouds also contributes importantly to the eastward extension and strengthening of the North Atlantic jet stream over Europe under global warming (Albern et al., 2021, 2019). Radiative effects of tropical clouds push the midlatitude eddy-driven jet equatorward, while those of extratropical clouds push it poleward (Watt-Meyer and Frierson, 2017). Biases in the Southern Hemisphere
jet location have also been traced back to too-weak shortwave reflection by clouds there (Ceppi et al., 2012). With regard to internal variability, anomalies in cloud-radiative effects can prolong the North Atlantic Oscillation and intensify the amplitude of the El Niño Southern Oscillation (Papavasileiou et al., 2020; Rädel et al., 2016). A more exhaustive description of these multifaceted cloud radiative-circulation couplings is provided by Voigt et al. (2020).

Constraining the cloud-radiative heating (CRH) profile is essential then to understand current-day circulation, as well as
its future changes with increased concentrations of atmospheric greenhouse gases. The vertical distribution of CRH, however, varies dramatically from one model to another and between models and satellite products (Cesana et al., 2019; Voigt et al., 2019). This variability is especially pronounced in the upper troposphere where ice clouds form and exists even amongst different reanalysis datasets (Tegtmeier et al., 2022). Our previous work has explored this variability in tropical upper-tropospheric CRH (Sullivan and Voigt, 2021; Sullivan et al., 2022). Structural differences in ice microphysics, such as consistency (or lack
thereof) in the treatment of ice crystal size or the initial size at which crystals are nucleated, are important drivers of CRH variability in storm-resolving simulations. High-resolution simulations also indicate that cloud macroproperties like degree of vertical overlap or decorrelation length between overlying cloud layers strongly influence radiative properties (Wang et al., 2021).

Wang et al. (2021) targeted tropical and Arctic mixed-phase clouds, and Sullivan and Voigt (2021) and Sullivan et al. (2022)
focused on tropical ice clouds because of the large intermodel CRH variability in these regions. Wang et al. (2021) note the influence of the width of the hydrometeor size distribution on CRH errors, while Sullivan and Voigt (2021) pinpoint several ice microphysical factors, such as initial ice crystal size and autoconversion rates, that drive CRH variability. Cesana et al. (2019) have compared heating rate profiles from several global climate models to CloudSat/CALIPSO data, and Hang et al. (2019) have produced a global climatology of radiative heating decomposed into cloud types from the CloudSat multisensor data
(CCCM). But sensitivities of midlatitude, atmospheric cloud-radiative heating to model settings remain relatively unexplored. An exception is the recent work of Senf et al. (2020), which found strong grid spacing dependence in shortwave top-of-atmosphere fluxes and a reduction in compensating longwave and shortwave biases at the finest grid spacings ($\sim 2.5$ km) over the North Atlantic. We extend this work on top-of-atmosphere fluxes to examine the in-atmosphere cloud-radiative heating.



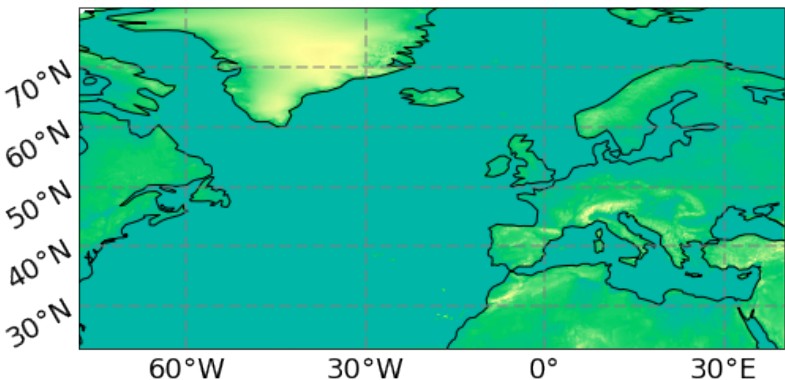

**Figure 1. The NAWDEX simulation domain covers the entirety of the North Atlantic as well as the northeastern Canadian seaboard, Greenland, and Europe.** The domain runs from 78°W to 40°E longitude and from 23°N to 80°N latitude.

We also build upon recent interest in the grid spacing and microphysics dependence of cloud-radiative heating, looking at how these model settings affect heating rates over the North Atlantic (e.g., Gettelman and Sherwood, 2016; Evans et al., 2017; Vannière et al., 2019; Sullivan et al., 2022). We start by establishing the climatological representativeness of our simulated cloud-radiative heating and present its dependencies on model settings, both in the net and decomposed into longwave and shortwave components. We examine whether these dependencies are due to different frequencies of specific cloud classes or whether the clouds in these classes have different properties. We then trace the changes in cloud class occurrence and condensate back to cloud-controlling factors. We close by identifying three model aspects at the root of the variability in North Atlantic cloud-radiative heating rates.

## 2 Methods

### 2.1 ICON Simulations

Simulations were performed with the Icosahedral Non-hydrostatic model (ICON) version 2.1.00 of the German Weather Service and Max-Planck Institute for Meteorology over a North Atlantic domain between 78°W and 40°E longitudinally and between 23°N and 80°N latitudinally (Fig. 1). We use the same set of simulations as presented in Senf et al. (2020). A brief description of these runs is presented here. After removing the spinup period, the ICON simulations extend over 14 days during the North Atlantic Waveguide and Downstream Impact Experiment (NAWDEX) field campaign: 21-25 and 30 September 2016, 1-5 October 2016, and 14-16 October 2016. NAWDEX was an international multi-aircraft field campaign taking place from 17 September to 22 October 2016 and based out of Iceland (Schäfler et al., 2018). NAWDEX studied midlatitude circulations, particularly warm conveyor belts, Rossby waves, and the North Atlantic jet stream, and the physical processes initiating and controlling them.



ICON is run during the NAWDEX period in numerical weather prediction mode with the convection scheme of Tiedtke (1989) updated by Bechtold et al. (2008) used at all grid spacings. For the simulations at 2.5 km grid spacing, the deep

convection scheme *or* both the deep and shallow convection schemes are switched off in order to investigate the effect of explicit treatment of convection. The impact of cloud microphysics is explored by switching between the one-moment microphysics of Doms et al. (2005) used in the operational NWP mode and the more sophisticated and computationally expensive two-moment microphysics of Seifert and Beheng (2006), where heterogeneous nucleation is prescribed as in Hande et al. (2015). Although the two-moment microphysics scheme was developed for convection-permitting resolutions, we use it here in combination

with parameterized convection also. For either the one- or two-moment scheme, the effective radius of cloud droplets or ice crystals is prescribed from the cloud liquid or ice water content respectively; this formulation makes microphysics and radiation inconsistent in the two-moment case (Kretzschmar et al., 2020). ICON uses the generalized cloud overlap scheme of Hogan and Illingworth (2000) and a diagnostic cloud cover scheme based upon a probability distribution of vapor mass mixing ratios relative to saturation (Giorgetta et al., 2018). The Rapid Radiative Transfer Model (RRTM) evaluates fluxes in our simulations

across 16 longwave and 14 shortwave spectral bands using a correlated-$k$ method (Mlawer et al., 1997).

Finally, six different horizontal grid spacings are used to encapsulate the range from typical global climate model meshes down to storm-resolving ones: 80, 40, 20, 10, 5, and 2.5 km. Across these grid spacings, the number of grid cells varies by three orders of magnitude. In the discussion below, the simulation with a grid spacing of $x$ km is sometimes referred to simply as the '$x$-km simulation'. Vertical grid spacing is held constant at 75 levels. Lateral boundary conditions with three-hourly

frequency and initial conditions come from the Integrated Forecast System. Surface and aerosol data come from the German Weather Service. We filter out grid points corresponding to land and sea ice from the NAWDEX domain in our results below, focusing only on cloud fields over ocean to remove differences due to surface albedo, surface temperature, or varying amounts of predicted sea ice.

## 2.2 Satellite, Reanalysis, and 'AMIP-like' Data

We compare our heating rate profiles to those from the 2B-FLXHR-LIDAR data, version P2R04 from CloudSat/CALIPSO data, binned to 2.5° resolution (see Papavasileiou et al. (2020)) and remapped to 0.25° resolution, over the North Atlantic domain during September and October between 2006 and 2011. As for the NAWDEX simulation output, we mask the land and sea ice grid points. Ice and liquid effective radii and water contents measured by the CloudSat cloud profiling radar and temperature and humidity profiles from the European Center for Medium-Range Weather Forecast (ECMWF) have been fed

to a two-stream radiative transfer model to compute 2B-FLXHR-LIDAR heating rates by L'Ecuyer et al. (2008). We also compare heating rates from the ERA5 reanalysis of the ECMWF to our ICON NAWDEX simulations (Hersbach et al., 2020). The ERA5 reanalysis assimilates radiances from both infrared sounders, such as AIRS and IASI, and geostationary satellites, such as GOES and Meteosat. Heating rates have then been generated within the reanalysis by applying RRTM and assumptions about ice crystal effective size and cloud condensation nuclei concentrations. We download these ERA5 heating rates at 0.25°

resolution over our domain from 2012 to 2016 in order to produce a climatologically representative profile.





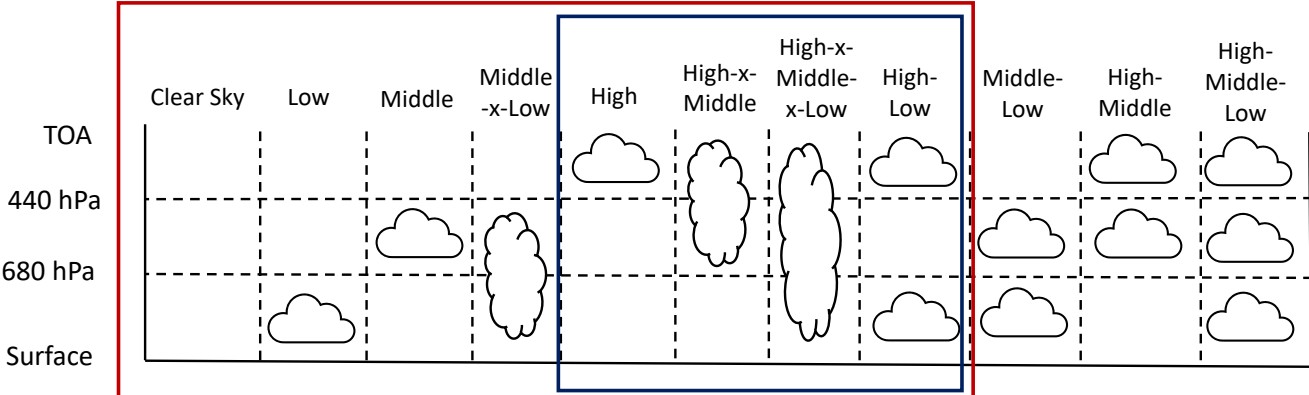

**Figure 2. The Cloud Vertical Structure classification of Oreopoulos et al. (2017) employs cloud fraction in three altitudinal ranges - Low, Middle, and High - to define 11 classes.** We use a subset of these shown in the red box and do not distinguish between continuous and discontinuous cloud layers. We also focus on upper tropospheric CRH influenced mostly by a smaller subset shown in the blue box. Adapted from Fig. 1 of Oreopoulos et al. (2017).

We also present CRH profiles from other coarse-resolution, 'AMIP-like' simulations with the ECHAM6 atmospheric component of the MPI-ESM model, the LMDz5A atmospheric component of the IPSL-CM5A model, and the ICON atmospheric model version 2.1.00 with a global R2B04 grid, corresponding to a horizontal grid spacing of approximately 160 km. These simulations employ climatological sea surface temperatures from the CMIP5 AMIP protocol and have been analyzed by Voigt

et al. (2019). Their CRH profiles are evaluated from over 5 or more years, so that we may interpret them as a North Atlantic climatology. In both the ICON NAWDEX and the 'AMIP-like' simulations, cloud-radiative heating is calculated as the difference between all-sky and clear-sky flux divergences.

### 2.3 Cloud Classes

Cloud layering strongly determines CRH, and decomposition of cloud fields into various Cloud Vertical Structure (CVS)

classes has proven useful in tracing the origins of atmospheric radiative warming and cooling (Oreopoulos et al., 2017; Lee et al., 2020). CVS classes build upon the International Satellite Cloud Climatology Project classification and are defined by cloud fraction thresholds at low (pressure ($p$) $\geq 680\,\mathrm{hPa}$), middle ($440\,\mathrm{hPa} \leq p \leq 680\,\mathrm{hPa}$), and high ($p \leq 440\,\mathrm{hPa}$) altitudes. Oreopoulos et al. (2017) define a classification consisting of High, Middle, Low, High-Middle, Middle-Low, High-Middle-Low, High-x-Middle, High-Low, Middle-x-Low, and High-x-Middle-x-Low clouds, as well as Clear Sky. *altitude 1-altitude*

*2* denotes cloudiness at altitudinal range 1 separated by clear sky from cloudiness at altitudinal range 2, whereas *altitude 1-x-altitude 2* denotes continuous cloudiness throughout altitudinal ranges 1 and 2.

Within the Low-Middle-High stratification, numerous possibilities exist when looking at the full cloud fraction field, as detailed in the Appendix of Oreopoulos et al. (2017). How many consecutive levels within an altitudinal range must have





cloud fractions greater than the threshold for the whole range to qualify as cloudy? Or if 20% of the cloud exists in the High
altitudinal range and 80% exists in the Middle altitudinal range, should it then be classified as isolated Middle or High-Middle?

We are mostly concerned with a general sensitivity of CRH to isolated versus deeper clouds, so we bypass some of these subtleties by employing a simplified version of the CVS classification with eight classes: isolated High, isolated Middle, isolated Low, High-x-Middle, Middle-x-Low, High-Low, High-x-Middle-x-Low, and Clear Sky (Fig. 2). To categorize cloudiness in a given grid cell, thresholds in cloud fraction are verified for the low ($p \leq 680$ hPa), middle ($440$ hPa $\leq p \leq 680$ hPa), and
high ($p \leq 440$ hPa) ranges. These two-dimensional low, middle, and high cloud fractions are calculated over the corresponding pressure ranges from the three-dimensional cloud fraction field using the generalized overlap assumption. If, for example, a column of grid cells has more than the threshold cloud fraction values in all three ranges, it is classified as High-x-Middle-x-Low. Or if it has only more than the threshold cloud fraction value in the low altitudinal range, it is classified as Low. We do not make the distinction between continuous and discontinuous layers of cloudiness. Three sets of thresholds were initially used,
based upon the following quantiles in the cloud fraction distribution: 60th-60th-25th, 62nd-67th-30rd, and 65th-70th-35th for high, middle, and low altitudinal ranges / cloud classes (Tab. S1). Our findings were neither qualitatively nor quantitatively dependent on these quantile-based thresholds (Fig. S1), and we show results from the intermediate set of thresholds.

## 2.4 Hackathon Format

The results presented here were generated in a non-traditional Hackathon format. Over the course of two years, our research
group met intermittently for intensive, 3-day periods of data analysis and discussion. Three subgroups focused on the climatological analysis (Sec. 3.1), the cloud class decomposition (Sec. 3.2), and the cloud-controlling factors (Sec. 3.4.2). This format facilitated communication about Python tools to handle the large datasets and a unique, group approach towards performing and organizing analyses.

## 3 Results

### 3.1 Climatological Cloud-Radiative Heating in the North Atlantic

Cloud-radiative heating (CRH) profiles averaged over open ocean in the NAWDEX domain from three global climate model simulations provide a first estimate of variability in North Atlantic climatological CRH (Fig. 3). The most prominent intermodel differences are in the lowermost ($p \geq 800$ hPa) and uppermost ($p \leq 300$ hPa) troposphere. The atmospheric component of the IPSL-CM5A model predicts by far the largest cloud-radiative cooling in the boundary layer and upper troposphere (maxima
of -2.2 K day$^{-1}$ and -1.1 K day$^{-1}$ respectively). These atmospheric coolings are more than five times the magnitude of those produced by the MPI-ESM model, while the CRH in ICON falls in between with larger boundary-layer cooling than MPI-ESM but smaller upper-tropospheric cooling. CRH profiles averaged over all longitudes between 23°N and 80°N mirror those over the NAWDEX domain, meaning that this midlatitude variability is not concentrated only over the North Atlantic. We also note





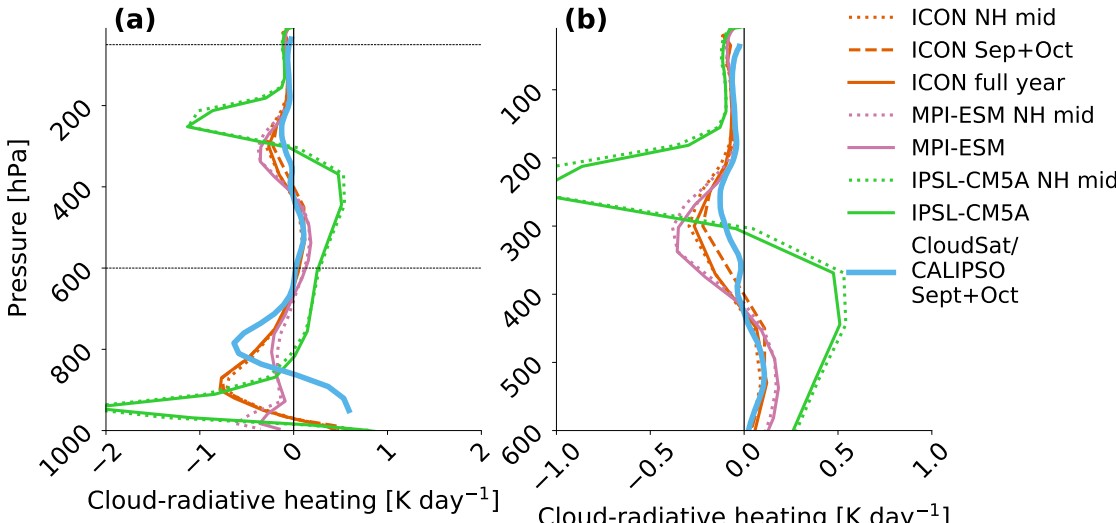

**Figure 3. North Atlantic climatological cloud-radiative heating varies five-fold in coarse-resolution global model simulations.** Full (panel a) and upper-tropospheric (panel b) CRH profiles averaged over the NAWDEX domain (23°N to 80°N and 78°W to 40°E) from the atmospheric components of the MPI-ESM, IPSL-CM5A, and ICON version 2.1.00 models, all with approximately 150-km horizontal grid spacing. The means between 23°N and 80°N over all longitudes for the three models are shown in the dotted traces denoted NH mid for Northern Hemisphere mid-latitudes. ICON profiles for both the full year *and* only September and October (Sep+Oct in the dashed trace) are shown. The dashed black lines in panel a indicate the subset of pressures shown in panel b.

that on the basis of the ICON simulations, September and October are representative months for the annually averaged North
Atlantic CRH (ICON full year *versus* ICON Sep+Oct).

Circulation effects of the differing CRH in these 'AMIP-like' simulations have been discussed by Voigt et al. (2019); their +4-K simulations show that particularly large CRH differences with warming are concentrated in the upper troposphere. The increase of upper tropospheric CRH with surface warming results in larger meridional temperature gradients and a poleward expansion of the Hadley cell and extratropical jets. If we can reliably construct the current-day upper tropospheric CRH, then
we also know what its profile should look like under global warming; clear-sky radiative cooling by water vapor provides a strong constraint for upper tropospheric cloud fraction and cloud top temperature globally (e.g., Thompson et al., 2017, 2019). Given the link of both current and future circulation to upper tropospheric CRH, we focus on the model dependencies above 5 km going forward.

We next examine the relative contribution of upper tropospheric CRH to the total, time mean, spatial mean heating rate within
our NAWDEX simulations (Fig. 4). This heating rate "climatology" for the North Atlantic is constructed from the simulations with coarsest grid spacing (80 km) and includes the longwave and shortwave cloudy and clear-sky radiative heating rates, as



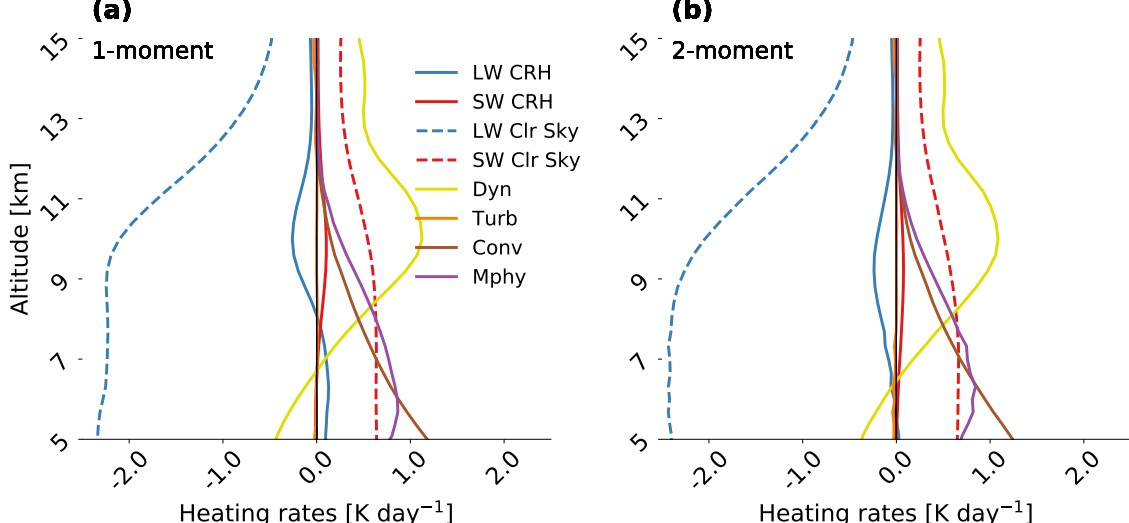

**Figure 4. The heating rate budget is dominated by clear-sky radiation and dynamics, but longwave cloud-radiative heating contributes non-negligibly in the upper troposphere.** Spatial mean, time mean vertical profiles of heating rate components at 80 km grid spacing in the one- (panel a) and two-moment (panel b) microphysics schemes. LW CRH is longwave cloud-radiative heating, SW CRH is shortwave cloud-radiative heating, LW Clr Sky is longwave clear-sky heating, SW Clr Sky is shortwave clear-sky heating, Dyn is dynamics, Turb is turbulence, Conv is convection, and Mphy is latent heating from microphysics and saturation adjustment.

well as dynamic, turbulent, convective, and microphysical heating rates:

$$\frac{\partial T}{\partial t} = \left(\frac{\partial T}{\partial t}\right)_{\text{CRH}} + \left(\frac{\partial T}{\partial t}\right)_{\text{Clr Sky}} + \left(\frac{\partial T}{\partial t}\right)_{\text{Dyn}} + \left(\frac{\partial T}{\partial t}\right)_{\text{Turb}} + \left(\frac{\partial T}{\partial t}\right)_{\text{Conv}} + \left(\frac{\partial T}{\partial t}\right)_{\text{Mphy}} \quad (1)$$

The largest component comes from clear-sky longwave radiative cooling (LW Clr Sky) followed by the dynamic heating (Dyn) and clear-sky shortwave radiative heating (SW Clr Sky). Thereafter, from about 9 up to 11 km, the microphysical heating and longwave cloud-radiative cooling are largest, with the latter contributing 14% to the overall budget.

The hierarchy and values of the heating rates are independent of whether we use a one- or two-moment microphysics scheme (Fig. 4a versus b). The longwave cloud-radiative heating profiles do differ qualitatively, however, in whether they exhibit an inflection point. While the longwave cloud component changes from cooling to heating around 7 km in the one-moment setup, it is exclusively cooling at the upper altitudes in the two-moment setup. These heating rates indicate that cloud-radiative heating, especially its longwave component, is non-negligible in the North Atlantic upper troposphere, and we further investigate its model dependencies.

We first construct net CRH profiles from our NAWDEX simulations across 6 horizontal grid spacings, with shallow convective parameterization only and explicit convection in the 2.5-km simulation, and using two different microphysics schemes (Fig. 5). Grid spacing dependence is subtle. Simulations with coarser grid spacing exhibit larger magnitude upper-tropospheric CRH, but profiles fall within one standard deviation of the 80-km profile over most of the upper troposphere. Then CRH





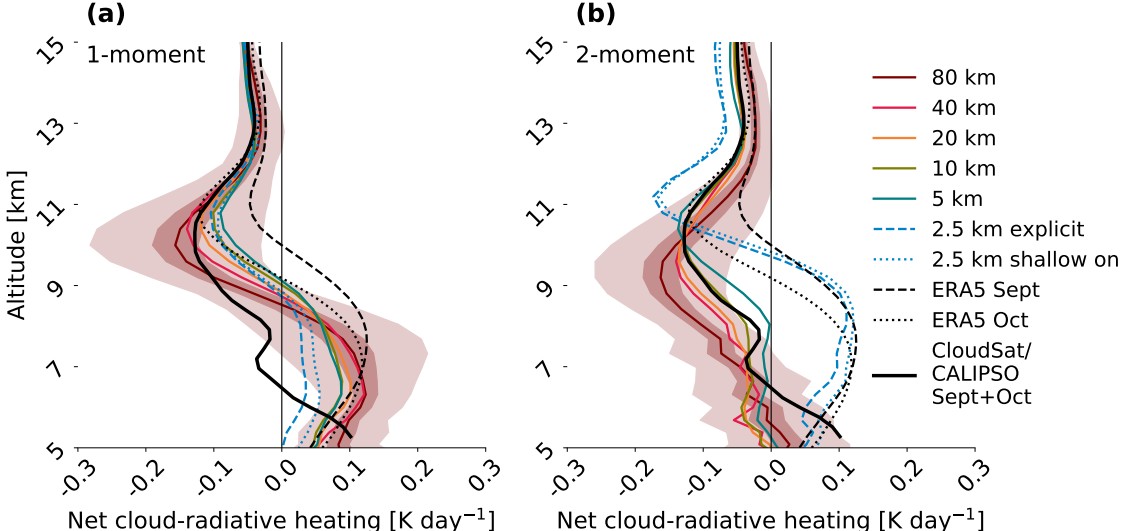

**Figure 5. Microphysics and convection dependency in the net CRH profile is much stronger than grid spacing dependency.** Upper-tropospheric, time mean, area mean net cloud-radiative heating from the ICON NAWDEX simulations at grid spacings from 2.5 km up to 80 km with a one- (panel a) and two-moment (panel b) microphysics scheme. The 2.5-km simulations either use only the shallow convection parameterization (shallow on) or explicitly represent both shallow and deep convection (explicit). The standard deviation and standard error over daily means are depicted as light and dark red shades atop the 80-km profile. Profiles from the ERA5 reanalysis in September (dashed black) and October (dotted black), as well as the CloudSat/CALIPSO 2B-FLXHR-LIDAR product (solid black), are also included.

changes qualitatively with the microphysics scheme from an S shape in the one-moment scheme (as in the 'AMIP-like' profiles of Fig. 3b) to a uniformly cooling profile in the two-moment scheme.

The most dramatic change occurs in turning off the deep convective parameterization in the two-moment microphysics simulations (Fig. 5b). Omitting the deep convective parameterization in the 2.5-km simulations shifts the upper tropospheric cooling peak upward by 2 km and narrows its vertical depth relative to the other simulations. Although these results are for the full simulation length in Fig. 5, they are robust for shorter durations down to a single day (Fig. S2).

Decomposing the net CRH into its longwave and shortwave components, we find that model dependencies are not isolated within a single component (Fig. 6). Both the longwave and shortwave CRH change more strongly with microphysics and convective scheme than with grid spacing. Interestingly, while the magnitude of longwave cooling increases at coarser grid spacing, that of shortwave heating decreases. Because longwave cooling is about twice as large as shortwave heating, it dominates the net CRH dependence. The larger spread on the longwave profiles also show that this component drives more of the CRH variability across days.

Atop the simulated CRH values—both net and decomposed into their longwave and shortwave components—we overlay ERA5 reanalysis values as well as a CloudSat/CALIPSO climatology, both over the NAWDEX domain during September and October. ERA5 assimilates observed radiances but still makes cloud microphysical assumptions within its radiative trans-

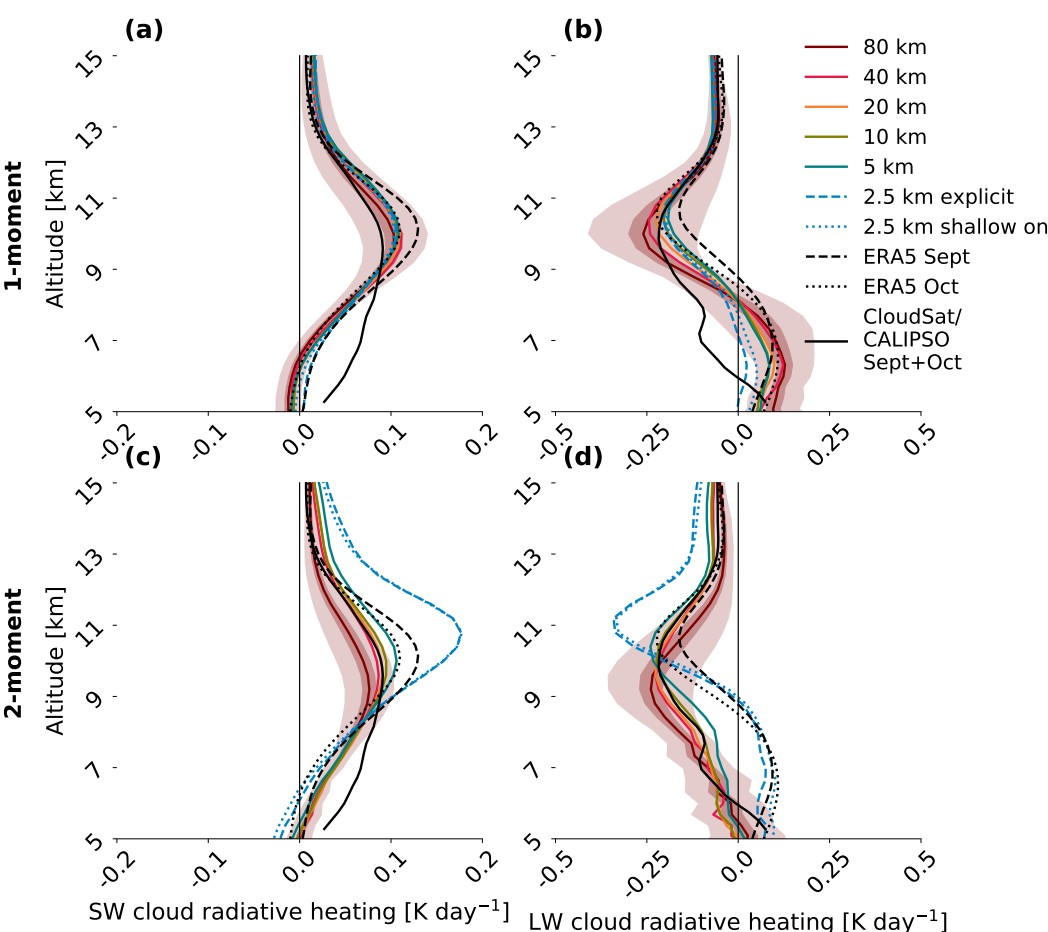

**Figure 6. Model setting dependency appears in both the longwave and shortwave components.** Upper-tropospheric, time mean, area mean shortwave (left panels) and longwave (right panels) cloud-radiative heating with all model settings as in Fig. 5. One- (top panels) and two-moment (bottom panels) microphysics schemes are shown, as well as profiles from the ERA5 reanalysis in September (dashed black) and October (dotted black) and the CloudSat/CALIPSO 2B-FLXHR-LIDAR product (solid black). Note the different x-axis limits on the left versus right panels.





fer calculations, along the lines of a one-moment scheme in which only cloud liquid and ice mass mixing ratios are tracked (e.g., Tiedtke, 1993; Forbes and Tompkins, 2011). The CloudSat/CALIPSO product (2B-FLXHR-LIDAR) incorporates cloud microphysical measurements into its calculation (Sec. 2.2). The ERA5 and CloudSat/CALIPSO profiles differ strongly from one another and from the simulations. The ERA5 profile has a muted version of the S-shape from the one-moment simu-

lations, whereas the CloudSat/CALIPSO profile shows uniform upper-tropospheric cooling by clouds as in the two-moment simulations.

Taking CloudSat/CALIPSO as our baseline, simulations with moderate grid spacing (10- or 20-km) and the two-moment microphysics compare most favorably. Using instead the ERA5 reanalysis as our baseline gives an indication of CRH with the cloud environment but not microphysics observationally constrained, and in this case, our simulations with the finest

grid spacing (2.5-km) and two-moment microphysics compare most favorably. None of the one-moment profiles mirror the CloudSat/CALIPSO or ERA5 profiles especially well. The messy state of this evaluation highlights a difficulty: Cloud-radiative heating is not directly observed, even from satellites, and associated radiative transfer or microphysical assumptions complicate any model-measurement comparison.

### 3.2 Cloud Class Decomposition

We turn next to understanding the strong convective and microphysical scheme dependency in the upper-tropospheric CRH by breaking it down into that associated with various cloud classes. Such a decomposition allows us to determine whether CRH differences are due to variations in heating associated with a particular cloud class *or* variations in the probability of occurrence associated with a particular cloud class. Stated mathematically, the total CRH is the summation over all such classes $i$ of the heating associated with a cloud class weighted by its frequency of occurrence ($f_i$ below):

$$\text{CRH} = \sum_i \text{CRH}_i \, f_i \qquad (2)$$

As detailed in Sec. 2.3, eight cloud classes are defined on the basis of cloud cover in three altitudinal ranges. Upper tropospheric CRH is driven primarily by four of these eight cloud classes: isolated High clouds, continuous High-x-Middle clouds, layered High-Low clouds, and deep High-x-Middle-x-Low clouds (blue box in Fig. 2). Physically, isolated high clouds correspond to cirrus formed in-situ or dissipating after formation as anvil outflow, whereas High-x-Middle-x-Low clouds represent forms of midlatitude deep convection, such as cyclones. The profiles associated with the Low, Middle, Middle-x-Low, and

Clear Sky regions are generally omitted, as these contribute negligibly to the CRH between 5 and 15 km (not shown).

Box plots of area-weighted occurrence frequency show negligible grid spacing dependence for all cloud classes (Figs. 7 and S3). For the classes including high clouds that are influential for upper tropospheric CRH, the mean occurrence changes less than 2% between the simulations with 80 and 2.5 km grid spacings. Otherwise, these box plots indicate that low clouds are the most frequent with a mean occurrence around 35%, followed by deep clouds (H-x-M-x-L) and clear sky with mean occurrences

of 20 and 18% respectively. Isolated middle clouds are least common followed by High-x-Middle clouds, occurring an average of 2 and 3% of the time respectively. Isolated high clouds also occur less frequently in this region with only 6% coverage on average.





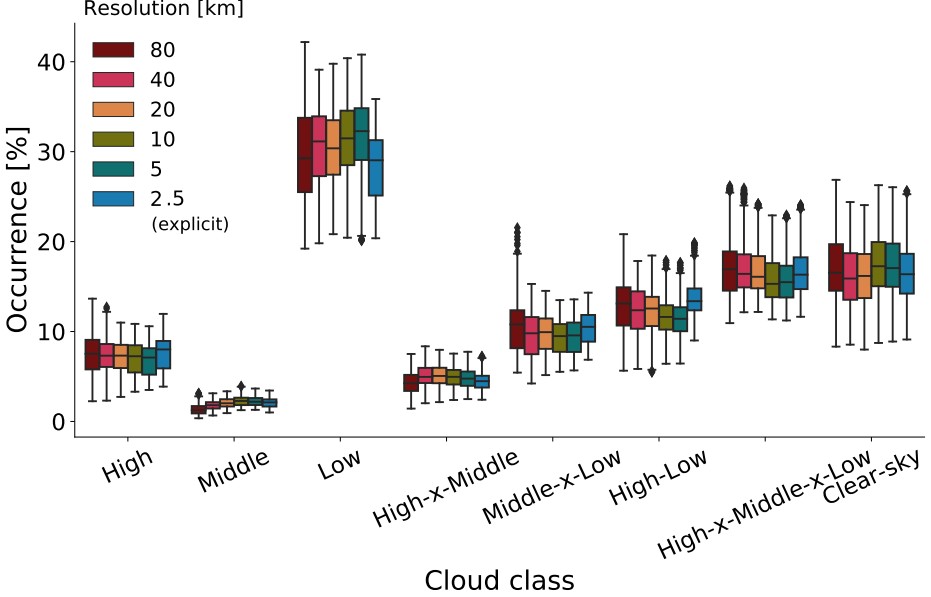

**Figure 7. There are no systematic changes in cloud class occurrence with grid spacing.** Area-weighted occurrence frequency for eight cloud classes across grid spacings for the simulations with two-moment microphysics. The box shows 25th (Q1), 50th (Q2), and 75th (Q3) percentiles. The whiskers show 1.5 times the interquartile range below the first quartile up to 1.5 times the interquartile range above the third quartile, i.e. $\left[\text{Q1-1.5(Q3-Q1)},\ \text{Q3+1.5(Q3-Q1)}\right]$. Diamonds indicate outliers. Fig. S3 is the same plot for the one-moment microphysics. Thresholds of the 62nd, 67th, and 30rd percentile of the cloud fraction distribution are used for high, middle, and low clouds, but mean occurrence is not sensitive to these thresholds (Fig. S1). The 2.5-km simulation uses neither a deep nor shallow convective parameterization (explicit). The sum of occurrence over all classes equals 1, and the sum over all classes except clear sky equals mean cloud fraction.

While the occurrence probabilities do not reflect the model dependencies of the net CRH, the cloud class filtered CRH does (Fig. 8). The isolated high clouds (High or High-Low) uniformly radiatively heat the upper troposphere between 5 and 15 km, whereas deeper clouds (High-x-Middle or High-x-Middle-x-Low) radiatively cool above about 8 km. For the isolated clouds, heating intensifies with finer grid spacing and especially turning off the convective parameterization with the two-moment scheme. In contrast, for the deeper clouds, cooling moderates with finer grid spacing. But again, the largest change in the radiative heating profile comes from turning off the convective parameterization with the two-moment scheme.

Having looked at both $f_i$ and $\text{CRH}_i$ from Eq. 2, we conclude that the latter factor drives the overall CRH dependencies. In other words, different model settings do not change the distribution of occurrence of various cloud classes, but only the CRH profiles associated with these cloud classes. Additionally, these changes are not limited to a single cloud class but rather appear across all of them containing high clouds.



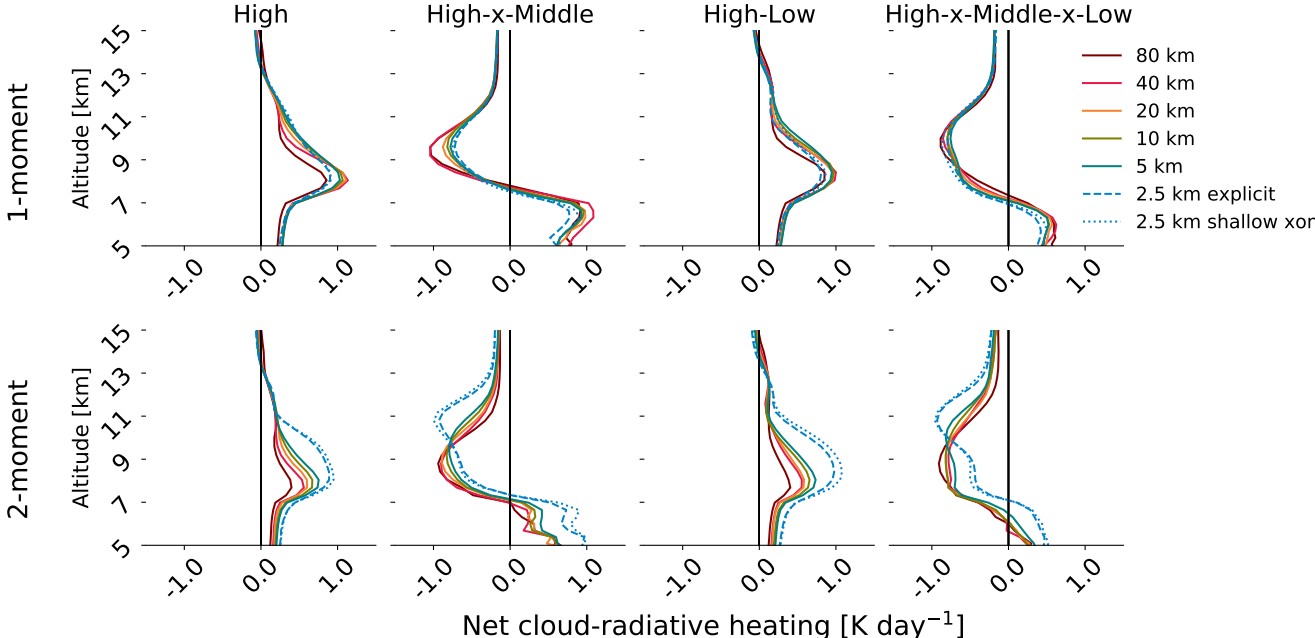

**Figure 8. Isolated high clouds heat and deep clouds cool the upper troposphere. All cloud classes containing high clouds contribute to the model dependencies in the CRH of the two-moment microphysics simulations.** Upper-tropospheric, time mean, area mean net cloud-radiative heating for four of the eight cloud classes with all model settings as in Fig. 5.

## 3.3 Cloud Properties by Class

We have ruled out varying occurrences of different cloud classes and now turn to cloud properties—overall and within the cloud classes—as an explanation for the model dependencies of CRH. An increased magnitude of time mean, area mean cloud-radiative cooling or heating can be due either to a larger amount of reflecting or absorbing condensate in the cloud, a greater coverage of the clouds, or both. We examine cloud liquid water ($q_c$), cloud fraction, and cloud ice mass mixing ratios ($q_i$) for the various simulation settings in Fig. 9. $q_c$ increases slightly with higher grid spacing in the two-moment scheme; however, its values are insufficient to drive the model dependencies in CRH (Fig. 9d).

Differences in cloud fraction qualitatively mirror those in CRH for the one-moment scheme (Fig. 9b): Cloud fraction peaks at a lower altitude and has a larger maximum in the simulations with coarser grid spacing, as does the cooling in its net CRH profiles. The correspondence of cloud fraction and net CRH dependence is weaker in the two-moment simulations (Fig. 9e). Cloud fraction is about 2% larger for the 2.5-km simulations, but otherwise there is no consistent trend with grid spacing or the altitude of maximum cloud fraction. This weak dependence of cloud fraction on model setting appears across the classes with high clouds (Fig. S4).

The primary driving factor of the large CRH changes with two-moment microphysics and explicit convection is then $q_i$ (Fig. 9f). The amount of cloud ice quadruples from about 5 mg kg$^{-1}$ in the 80-km simulation to about 19 mg kg$^{-1}$ in the two





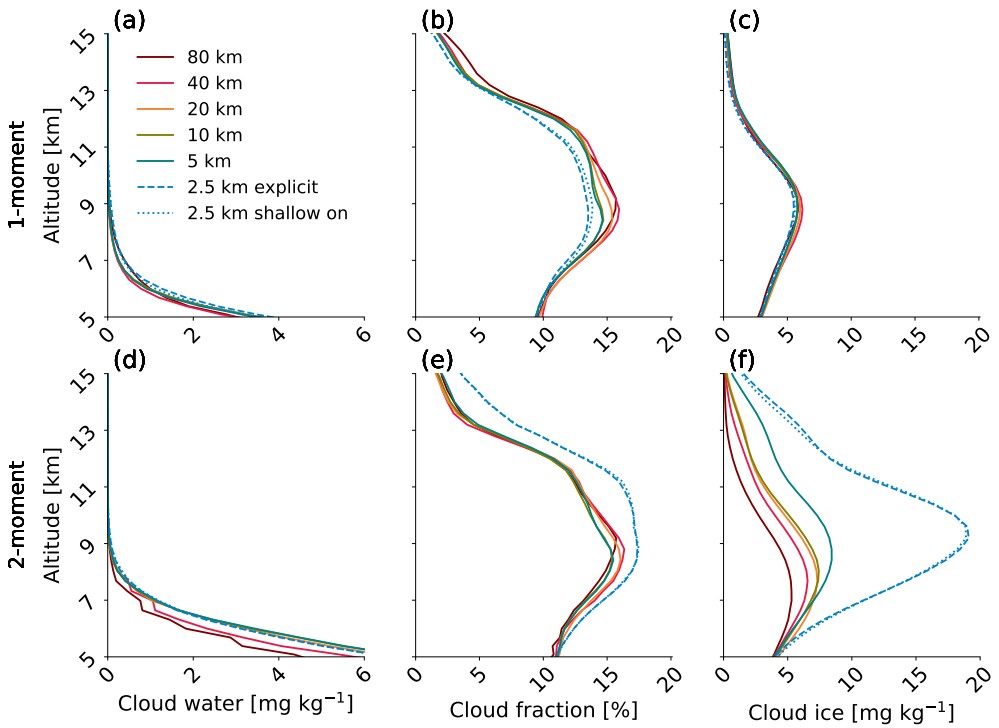

**Figure 9. Changes in the cloud ice mass mixing ratio drive the model dependencies of upper-tropospheric CRH in the two-moment simulations.** Upper-tropospheric, time mean, area mean profiles of cloud water mass mixing ratio (panels a and d), cloud fraction (panels b and e), and cloud ice mass mixing ratio (panels c and f) for the one- (top panels) and two-moment (bottom panels) microphysics simulations with all model settings as in Fig. 5.

2.5-km simulations (without shallow or any convective parameterization). The one-moment simulations show no such change in $q_i$ with model settings (Fig. 9c). As in Sec. 3.2, we can decompose these $q_i$ differences into those associated with various
cloud classes. Fig. 10 illustrates that the $q_i$ increases with grid spacing are somewhat larger for the deeper cloud layers—the High-x-Middle and High-x-Middle-x-Low classes—than for the isolated high clouds but occur qualitatively across all the classes with high clouds. Likewise, the lack of grid spacing and convection dependence in $q_i$ for the one-moment schemes is uniform across classes; there are no compensating differences in $q_i$.

The model uses only condensate mass to calculate CRH. However, CRH is also physically determined by hydrometeor
number, and we examine cloud ice crystal numbers ($N_i$) from our simulations to understand how their omission may affect CRH. $N_i$ profiles parallel $q_i$ ones for the two-moment microphysics simulations (Fig. 11, top panels). The runs without a deep convective parameterization produce more than four times as many ice crystals as those with a convective parameterization. Not only is more ice mass produced in the clouds, it is distributed over many more hydrometeors. In physically accurate frameworks, larger $N_i$ should promote multiple scattering and eventual absorption of solar radiation, enhancing the shortwave
heating peak (Fig. 6c). Distribution of ice mass over many more crystals could also prolong cloud lifetime and enhance CRH.



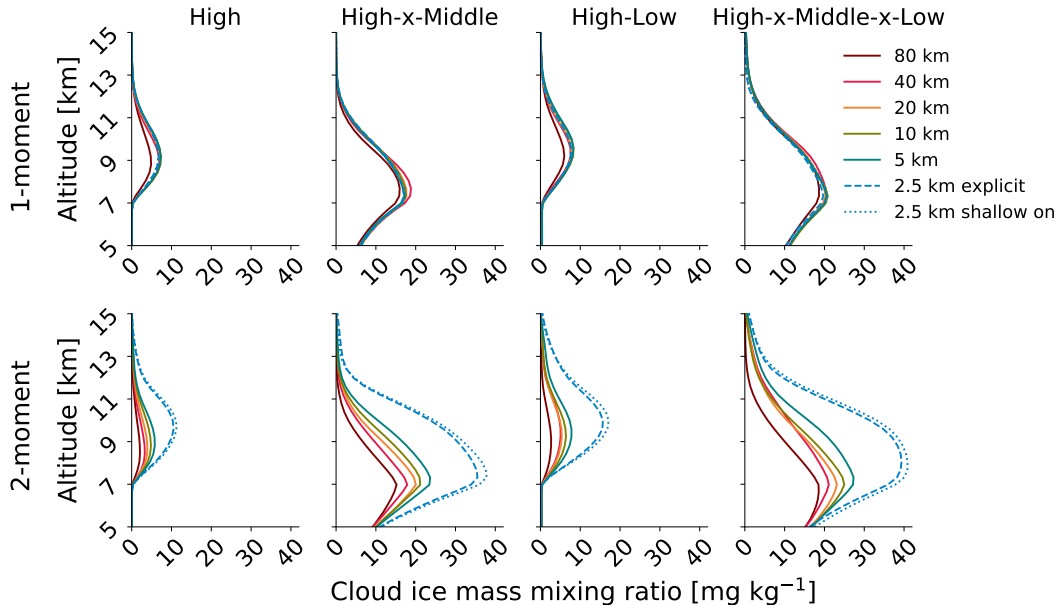

**Figure 10. Cloud ice mass mixing ratio increases four-fold from the coarsest to finest grid spacing simulations.** Diagnostic ice mass mixing ratios from one- (top panels) and two-moment (bottom panels) simulations for the four cloud classes that include high clouds with all model settings as in Fig. 5.

Our simulations permit such a cloud lifetime effect insofar as it is independent of CRH, but the cloud occurrence and cloud fraction results above indicate that it is not dominant.

Along with liquid and ice crystals, upper tropospheric clouds may also contain snow ($q_s$) and graupel ($q_g$). Whereas $q_i$ showed no model dependency for the one-moment simulations, the maximum in $q_s$ changes fourfold from the 80-km simulation

down to the 2.5-km one without convective parameterization (Fig. 11, bottom). This monotonic increase in $q_s$ appears for all cloud classes, and especially the deep clouds with the one-moment scheme. Similarly, the $q_g$ maximum changes by an order of magnitude across these model settings between 5 and 15 km (Fig. S5). It is important to note that snow and graupel do not interact with the radiative transfer scheme in ICON. This exclusion of certain hydrometeors from the radiation scheme is motivated in part by size and in part by lack of a fractional coverage variable (e.g., Xu and Randall, 1995). Graupel will tend

to sediment out more rapidly than the time step used to call the radiation scheme, whereas the fractional coverage of snow, distinct from the liquid or ice cloud fraction, is not a tracked variable. We can therefore conclude that grid spacing dependence for the one-moment microphysics is concentrated in radiatively inactive cloud species.

### 3.4 Understanding Cloud Property Differences

As a final step, we ask why the High, High-x-Middle, High-Low, and High-x-Middle-x-Low clouds produce more ice and have

slightly higher coverage in the two-moment simulations. We have advocated in our work on tropical cloud-radiative heating for



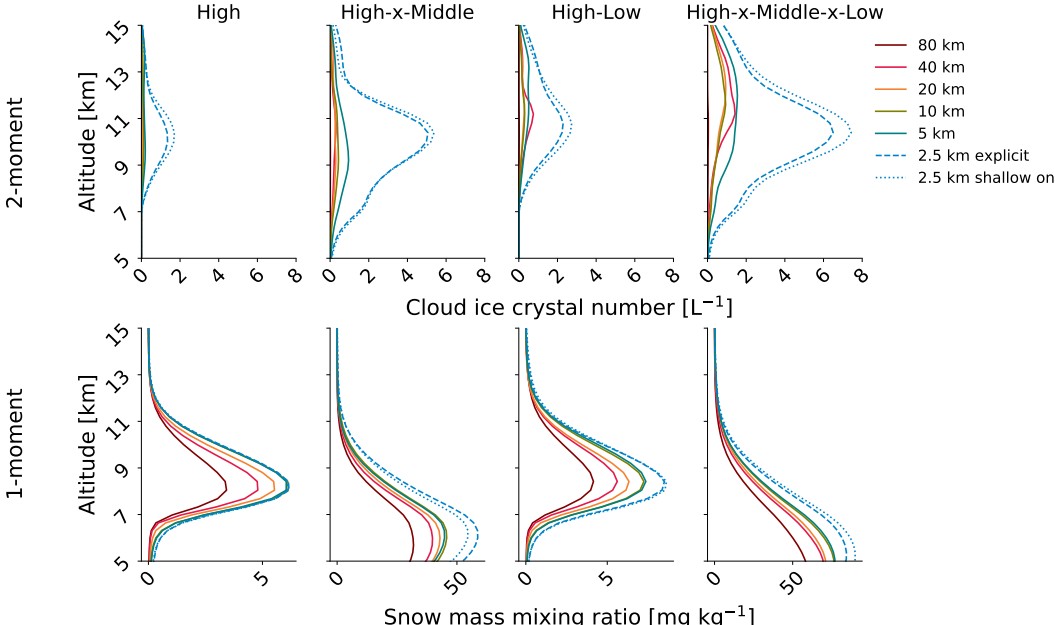

**Figure 11. Strong grid spacing dependence appears in both the ice crystal numbers from the two-moment scheme and the snow mass mixing ratios from the one-moment scheme.** Cloud ice crystal number from the two-moment simulations (top panels) and snow mass mixing ratios from the one-moment simulations (bottom panels) for the four cloud classes that include high clouds with all model settings as in Fig. 5.

process decomposition as a means of unraveling such differences (Sullivan and Voigt, 2021; Sullivan et al., 2022). This process decomposition can be done in a number of ways. Processes can be classified based upon the temperature range in which they are active to generate an "altitudinally stratified recipe" for CRH (Sullivan and Voigt, 2021). Processes can also be organized based upon when they occur within the cloud lifecycle to produce a "temporally stratified recipe " for CRH (Sullivan et al., 2022).

Here, processes are categorized as sources versus sinks of cloud ice. Then differences are understood either by structural differences in the source-sink formulations *or* by differences in the inputs to these formulations. Stated mathematically:

$$(q_i, N_i) = \phi(\text{CPs}, \text{CCFs}) - \psi(\text{CPs}, \text{CCFs}) \tag{3}$$

where $\phi$ and $\psi$ represent microphysical sources and sinks respectively, CP denotes a cloud parameter like the deposition density of ice crystals, and CCF denotes cloud-controlling factors, a term for the environmental conditions that determine cloud properties (e.g., Stevens and Brenguier, 2009).

Within the two ice microphysics schemes in ICON, ice mass can be consumed by autoconversion, melting, and sedimentation. Because $q_c$ differences are so much smaller than those in $q_i$, we focus on sink processes that do not involve the liquid phase: autoconversion and sedimentation. Ice mass can also be generated by nucleation, droplet freezing, depositional growth,



and riming. Somewhat larger cloud water mixing ratios at finer grid spacing in the two-moment simulations may contribute to
slightly stronger riming and droplet freezing tendencies (Fig. 9). However, these processes cannot be the primary driver for the
$q_i$ differences of much larger magnitude. We focus instead on nucleation and growth sources.

### 3.4.1 Cloud Ice Sources and Sinks

Autoconversion is the process converting between ice and snow, with its rate $S_{\text{auc}}$ represented as follows in the two microphysics
schemes:

$$S_{\text{auc, 1M}} = (10^3 \, s^{-1})(q_i - q_{i,0}) \tag{4}$$

$$S_{\text{auc, 2M}} = E_{ii} N_i q_i G(\delta_i, \theta_i) \tag{5}$$

where $q_{i,0}$ is a threshold ice mass mixing ratio before autoconversion initiates, set to 0 in the one-moment scheme; $E_{ii}$ is the
ice-ice collision efficiency; and $G$ is a function of $\delta_i$ and $\theta_i$, non-dimensional combinations of gamma distribution parameters
representing the ice crystal sizes. The one-moment formulation simply transfers ice to snow over a fixed time constant. This
sink is then much stronger than in the two-moment formulation, which incorporates dependence on the crystal numbers and
relative sizes.

Snow and ice settle at the following terminal velocities in the one- and two-moment schemes:

$$v_{Ts,1M} = (7.37 \, \text{m s}^{-1} \, \text{kg}^{-0.125}) \, m_s^{0.125} \tag{6}$$

$$v_{Ts,2M} = (8.156 \, \text{m s}^{-1} \, \text{kg}^{-0.526}) \, m_s^{0.526} \tag{7}$$

$$v_{Ti,2M} = (317 \, \text{m s}^{-1} \, \text{kg}^{-0.363}) \, m_i^{0.363} \tag{8}$$

where $m_s$ is the snow crystal mass and $m_i$ is the ice crystal mass. Ice does not sediment in the one-moment scheme. For a range
of hydrometeor masses $\sim \mathcal{O}(10^{-13} \, \text{kg}$ up to $10^{-10} \, \text{kg})$, the terminal settling velocity for snow in the one-moment scheme is
much stronger than that for either ice or snow in the two-moment scheme. The sedimentation sink then is also much stronger
in the one-moment formulation.

Heterogeneous nucleation occurs on ice-nucleating particles (INP), represented as follows in the one- and two-moment
schemes respectively:

$$C_{\text{INP, 1M}} = (1 \times 10^2) \exp\left[-0.2(T - 273 \, \text{K})\right] \tag{9}$$

$$C_{\text{INP, 2M}} = \begin{cases} (4.99 \times 10^4) \exp\left[-0.2622(T - 237 \, \text{K})^{1.2044}\right] & \text{(10)} \\ (7.72 \times 10^4) \exp\left[-0.0406(T - 220 \, \text{K})^{1.4705}\right] f(\text{RH}_{\text{ice}}) & \text{(11)} \end{cases}$$

where $T$ is subzero temperature and $\text{RH}_{\text{ice}}$ is the relative humidity with respect to ice. While the one-moment scheme repre-
sents only immersion nucleation (Eq. 9), the two-moment scheme represents both a relative humidity-dependent deposition
nucleation and immersion nucleation (Eqs. 10 and 11 respectively). Both formulations predict exponential increases in INP
as subzero temperature cools, but with a much steeper slope in the two-moment than one-moment scheme. Conversely, the
absolute INP number from the one-moment scheme is much higher (e.g., Sullivan et al., 2022, Figure 10a).





Finally, the rate of depositional growth $S_{\text{dep}}$ is represented with a much more complicated temperature dependence in the two-moment scheme:

$$S_{\text{dep, 1M}} = (1.3 \times 10^{-5}) m_i^{1/3} (q_v - q_{\text{sat,i}}) \tag{12}$$

$$S_{\text{dep, 2M}} = \frac{4\pi C_i D_i S_i f(m_i)}{\left[ \frac{RT}{p_{\text{sat},i}\mathcal{D}} + \frac{L_{iv}}{k_i T}\left(\frac{L_{iv}}{RT} - 1\right)\right]} \tag{13}$$

where $q_v$ is the specific humidity; $q_{\text{sat},i}$ and $p_{\text{sat},i}$ are the saturation specific humidity and vapor pressure with respect to ice; $C_i$ is crystal capacitance; $f(m_i)$ represents a mass-dependent ventilation coefficient; $k_i$ is the thermal conductivity of ice; $L_{iv}$ is the latent heat of sublimation; $D_i$ is diffusivity of vapor water; $S_i$ is the saturation with respect to ice; and $R$ is the gas constant. Key to both the nucleation and growth sources is the initial mass at which ice crystals are formed. The two-moment scheme initiates its crystals at $10^{-14}$ kg, and the one-moment scheme at a much larger mass of $10^{-12}$ kg (e.g., Sullivan et al., 2022,

Table 2). While the two-moment scheme generates fewer smaller crystals, they also stay aloft longer.

### 3.4.2 Cloud Controlling Factors by Class

Looking at the cloud ice source and sink formulations above, temperature ($T$), specific humidity ($q_v$), and supersaturation generation represented by vertical velocity ($w$) are the most important cloud-controlling factors (CCFs). We examine these inputs across cloud classes and model settings (Fig. 12). Specific humidity differences from the 80-km simulation are quite

small (Fig. 12, top row). The simulations with finer grid spacing are drier than the 80-km one below 10 km, but there is not a smooth trend toward lower specific humidity with finer grid spacing.

Profiles of temperature difference from the 80-km simulation mostly indicate a consistent trend of upper tropospheric temperatures cooling as grid spacing is refined, aside from the 2.5-km simulations (Fig. 12, middle row). Across all classes with high clouds, the 40-km simulation is about 0.5 K cooler than the 80-km one between 5 and 11 km; the 5-km simulation is as

much as 1.8 K cooler at these altitudes. These shifts toward colder temperatures below 11 km can help explain the increasing $q_i$ there at finer grid spacings. Colder temperatures will accelerate nucleation of new crystals and depositional growth of existing crystals in the two-moment scheme. However, the trend does not hold for the 2.5-km simulations without convective parameterization. Variations in input temperature cannot explain the dramatic increase in $q_i$ with explicit convection.

Vertical velocities increase systematically with refined grid spacing, especially for the deep cloud layers (Fig. 12, bottom

row). Deep cloud layers—High-x-Middle and High-x-Middle-x-Low classes—are characterized by ascent throughout, whereas the isolated cirrus—High or High-Low classes—have ascending air only above 7 km with descent below. They will also promote nucleation and growth in the same manner as temperature. For the High-x-Middle clouds, vertical velocity increases by a factor of 1.8 from 1.2 m s$^{-1}$ at 80-km grid spacing up to 2.2 m s$^{-1}$ at 2.5-km grid spacing. And for the High-x-Middle-x-Low clouds, vertical velocity increases by a factor of 1.4 from 2.5 m s$^{-1}$ at 80-km grid spacing up to 3.5 m s$^{-1}$ at 2.5-km

grid spacing.

A subtlety of vertical velocity is that a few instances of strong ascent can drive the majority of ice nucleation (e.g., Donner et al., 2016; Sullivan et al., 2016; Shi and Liu, 2016). The extreme values are more influential than the means depicted in Fig.



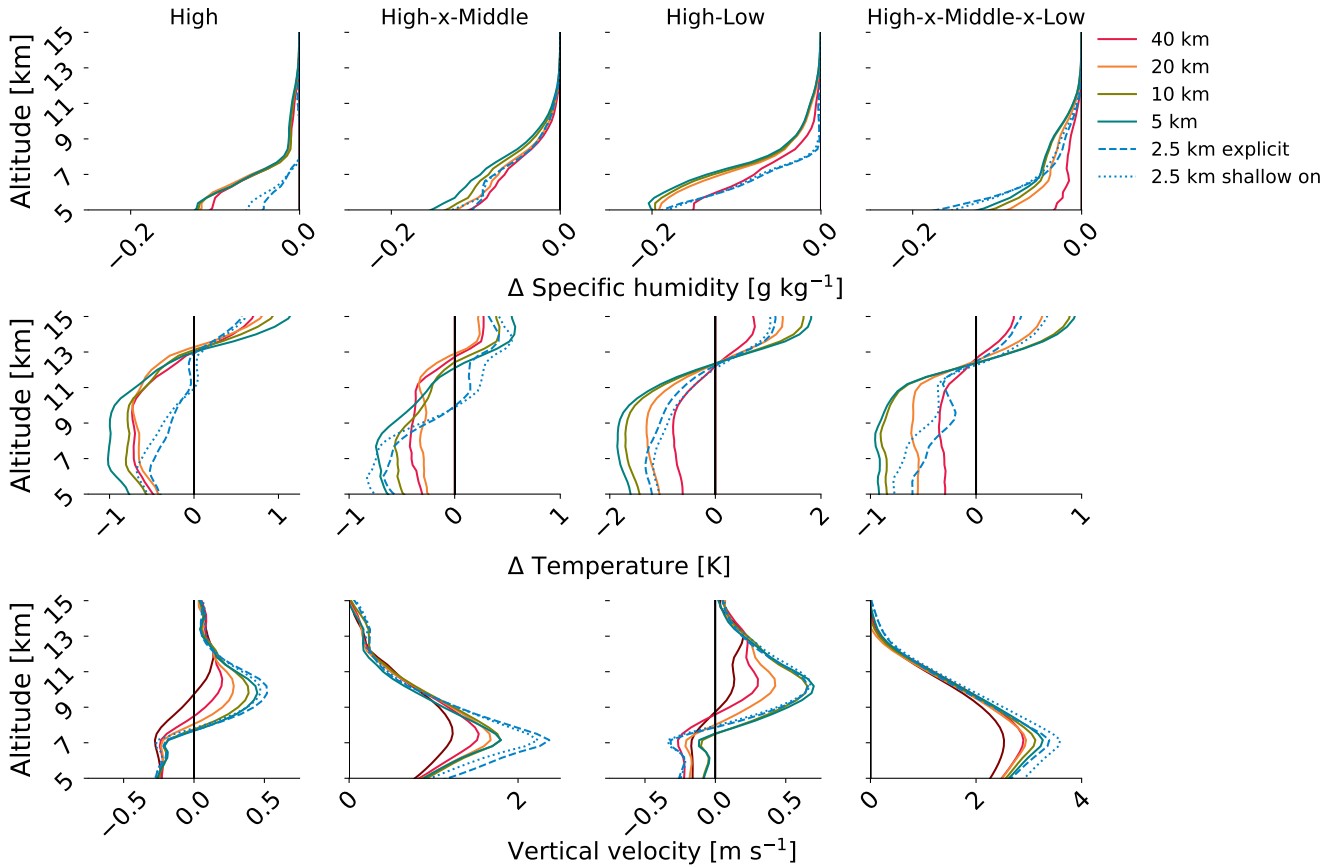

**Figure 12. Simulations without convective parameterization exhibit stronger mean vertical velocities.** Differences in the upper-tropospheric, time mean, area mean specific humidity (top row) and temperature profiles (middle row) from that of the 80-km simulation. Time mean, area mean vertical velocity profiles for all simulation settings (bottom row). Variables associated with the four cloud classes that include high clouds are shown for the simulations with the two-moment scheme only with all model settings as in Fig. 5.

12, so we also construct the probability distribution of vertical velocities at 500 hPa from the various simulations (Fig. 13). The tails of these vertical velocity distributions become fatter for finer grid spacing and without convective parameterization 350 for both the one- and two-moment microphysics schemes. This distribution broadening indicates that vertical velocities, not only in the mean but also in the extremes, intensify at higher grid spacings.

A final factor to consider is coupling of the microphysics and convection schemes. When convection is parameterized, only cloud mass mixing ratio, not hydrometeor number, is passed between the two schemes, as seen in the buoyancy equation (e.g.,



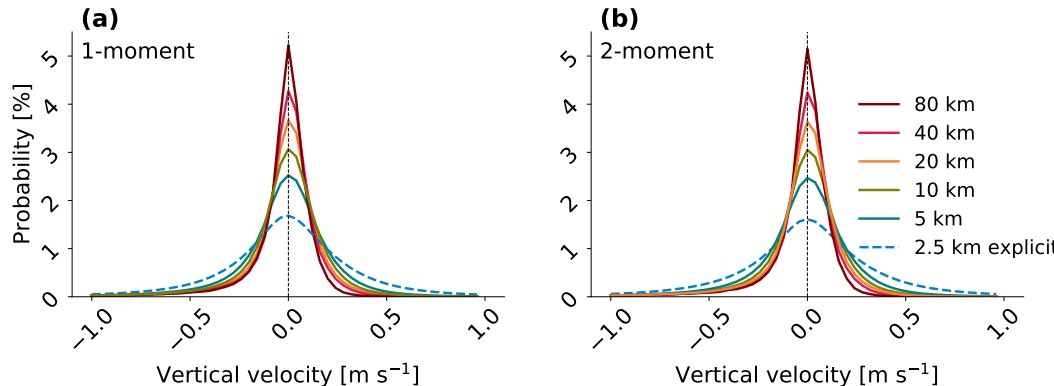

**Figure 13. Instances of the strongest ascent and descent at 500** hPa **both become more probable at higher grid spacings.** Probability distributions of vertical velocity from the simulations with the one-moment (panel a) and two-moment microphysics (panel b) with all model settings as in Fig. 5.

Möbis and Stevens, 2012):

$$B = g \left[ \frac{T_{\mathrm{cld}} - T_{\mathrm{env}}}{T_{\mathrm{env}}} + \varepsilon_R(q_{v,\mathrm{cld}} - q_{v,\mathrm{env}}) - (q_c + q_i) \right] \quad (14)$$

where the subscript cld denotes cloud, the subscript env denotes the environment, $\varepsilon_R$ is the ratio of moist and dry air gas constants minus one, and $g$ is gravitational acceleration. The more extreme vertical velocities and much higher $N_i$ in the 2.5-km simulations without convective parameterization speak in favor of a convective invigoration whereby more cloud condensate intensifies ascent via diabatic heating and vice versa. While contradictory results have been produced in the literature on convective invigoration, several recent studies find positive relationships between vertical velocity and condensed water in

deep convection (Abbott and Cronin, 2021; Marinescu et al., 2021; Grant et al., 2022).

This analysis of source and sink processes and the cloud-controlling factors driving them produces a balance in favor of larger ice production within the two-moment scheme, and especially with explicit convection. The most important elements in this balance are 1) weaker autoconversion and sedimentation sinks; 2) smaller initial crystal sizes; and 3) more instances of strong vertical velocity in the two-moment setup with explicit convection.

**4   Conclusions**

Given the importance of cloud-radiative heating—especially its upper-tropospheric portion—to large-scale circulation features from the Hadley circulation to the eddy-driven jet, we have explored its dependencies on grid spacing, convective parameterization, and microphysics scheme in a numerical weather prediction model. The combination of parameterized versus explicit representation of convection *and* a one- versus two-moment microphysics scheme is the most influential model setting for CRH

in our simulations. When we use a two-moment microphysics scheme, switching from parameterized to explicit convection has a much more dramatic effect than in the one-moment simulations. We posit that an inconsistent microphysics-convection



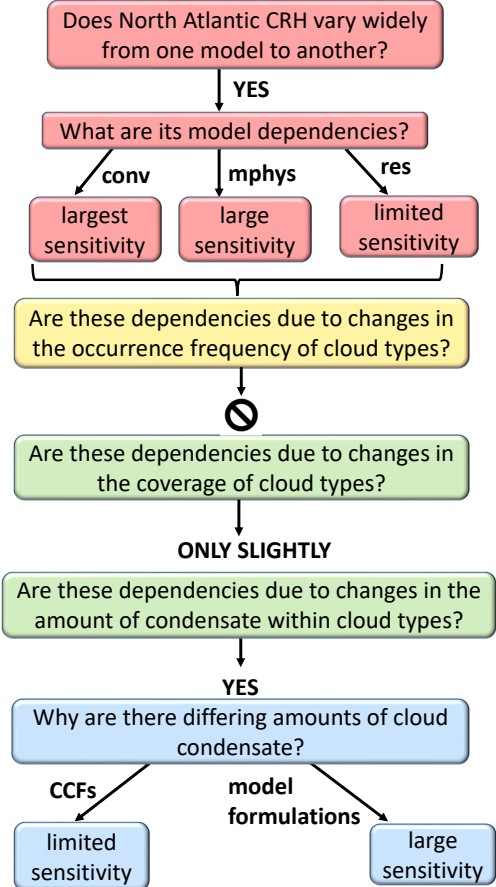

**Figure 14.** A schematic overview of our analysis. Results of Secs. 3.1, 3.2, 3.3, and 3.4 are shown in pink, yellow, green, and blue respectively.

coupling, one in which only ice mass mixing ratio not ice crystal number is used to evaluate buoyancy, is at the root of this sensitivity. Sensitivity to grid spacing is more muted than those to the microphysics or convection parameterizations (Fig. 14). This result reflects the increased importance of constraining microphysical uncertainties as we transition toward the higher grid

spacings of storm-resolving models.

Strong microphysical and convective sensitivity and weaker grid spacing sensitivity in the CRH profiles do not appear in distributions of cloud class occurrence and appear only weakly in cloud fraction profiles. Instead, it is the cloud ice mass mixing ratio profiles that mirror the CRH dependencies most closely. We can trace these cloud ice mass mixing ratio differences back one additional step to changes in microphysical formulations and cloud-controlling factors (Fig. 14). Radiatively inactive frozen

species, like snow and graupel, and the initial ice crystal mass, via its effect on subsequent growth and sedimentation rates, are two influential aspects of the microphysical formulations. Within the cloud-controlling factors, the width of the vertical velocity distribution, as well as upper-tropospheric temperature, vary systematically with model setting.



Importantly, these findings are robust to several factors. The dependencies affect both shortwave and longwave compo-nents of the cloud-radiative heating and occur across isolated cirrus, layered cirrus-bounday layer cumulus, and forms of deep

convection (High, High-Low, High-x-Middle, and High-x-Middle-x-Low in our decomposition). They are also not dependent on the cloud fraction thresholds used to define these cloud classes (Fig. S1) or on the simulation duration. The grid spacing and scheme dependencies already emerge within a single-day simulation (Fig. S2). The upper-tropospheric CRH variability motivating this work also appears not only across three coarse-resolution global climate models (Fig. 3) but also across four reanalysis datasets (Tegtmeier et al., 2022) and between ERA5 reanalysis and the CloudSat/CALIPSO 2B-FLXHR-LIDAR

data (Fig. 5).

This last point highlights a challenge in further constraining atmospheric cloud-radiative heating: Even our baseline contains uncertainties or assumptions. The disagreement between the ERA5 and CloudSat/CALIPSO profiles indicates that thermody-namic and wind fields are insufficient to constrain CRH. Both the one- and two-moment microphysics scheme generate quite similar distributions of cloud class occurrence despite drastically different upper-tropospheric CRH profiles (Fig. 7 and Fig.

S3). Stated another way, both cloud macrophysical and microphysical properties are needed to predict CRH. This result echoes our previous work on tropical CRH: Four-fold CRH variability can be produced by "flipping ice microphysical switches" in the model (Sullivan and Voigt, 2021), and cloud ice water content still changes five-fold when inputs to microphysics schemes are fixed via a "Lagrangian piggybacking" technique (Sullivan et al., 2022).

A second challenge is that different combinations of model settings may improve model-measurement agreement in top-

of-atmosphere or surface radiative fluxes versus atmospheric cloud-radiative heating. Models are often tuned based upon their outgoing longwave radiation, but as we have noted throughout, it is the in-atmosphere heating that feeds back upon circulation. As an example, Senf et al. (2020) use the same set of simulations as we do to assess the top-of-atmosphere cloud-radiative flux. They find the best agreement with the CloudSat/CALIPSO climatological value from the simulation with 2.5-km grid spacing and shallow convective parameterization only. In our NAWDEX simulations, it is instead more moderate grid spacings and

parameterized convection that agree best with the CloudSat/CALIPSO CRH (Fig. 5).

One suggestion for progress is to use atmospheric measurements to study atmospheric heating. This study and almost all other existing work use top-of-atmosphere (satellite) measurements to assess and investigate simulated atmospheric CRH. Although these satellite data have much better coverage and provide more robust statistics, it seems natural to use in-situ radiative flux and cloud microphysical measurements to further investigate the in-atmosphere link of CRH to small-scale cloud

properties. Our future work will adopt this approach in looking back at CRH over the Asian monsoon region.

*Code and data availability.*  All codes to reproduce figures from model output are available at https://github.com/sylviasullivan/nawdex-hackathon and postprocessed data is available in an online repository at https://zenodo.org/record/7236564 Sullivan (2022). No proprietary software has been used in this work. The ICON model source code is availabe upon request to icon@dwd.de. Postprocessing was performed with Jupyter Notebooks.



*Author contributions.*   All authors contributed to conceptualization, formal analysis, and visualization. AV led funding acquisition and project administration, and SS led writing of the original draft. All authors contributed to review and editing of the manuscript draft.

*Competing interests.*   The authors declare that they have no competing interests.

*Acknowledgements.*   All authors were supported by the German Ministry of Education and Research (BMBF) and FONA: Research for Sustainable Development (www.fona.de) under grant 01LK1509A. SS and AV acknowledge the larger group of researchers and students in
the NSF-PIRE project 1743753 whose ideas have contributed to this work. We thank also Blaž Gasparini and Axel Seifert for discussions regarding the microphysics-convection coupling in the ICON model and the German Climate Computing Center (DKRZ, Hamburg) for computing and storage resources as part of project 1018.



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
