# Peer review of "How Does Cloud-Radiative Heating over the North Atlantic Change with Grid Spacing, Convective Parameterization, and Microphysics Scheme \*in ICON version 2.1.00?"

_EGUsphere, 2023_

## Author Comment (AC1)

**Reviewer and Editor Responses: "How Does Cloud-Radiative Heating over the North Atlantic Change with Grid Spacing, Convective Parameterization, and Microphysics Scheme?"**

**Reviewer 1**
This study explores the sensitivity of model representation of atmospheric cloud radiative heating profiles over the North Atlantic to changes in grid resolution, atmospheric convection (explicit or parameterized), and microphysics scheme within the ICON model. While grid resolution is only found to play a small role, cloud radiative heating profiles are highly sensitive to the model representation of convection and microphysics. In particular, the role of cloud ice mass mixing ratio appears to play a critical role.

This manuscript provides a good discussion of the factors governing atmospheric cloud radiative heating profiles at midlatitudes in the ICON model. Technically, the manuscript is sound and just needs some minor corrections/clarifications (detailed below). However, my general impression in reading this paper is that I'm not sure GMD is really the right journal for this work. The manuscript is using the model output data set from a previous study (Senf et al. 2020) and really not describing fundamentally new methods, rather than just extending the authors' previous work from the tropics to the midlatitudes. I'll leave it to the editor to decide whether GMD is the appropriate venue for this work.

We thank the reviewer for their time and effort in evaluating our work and for their feedback. We also recognize the reservation about GMD as the appropriate journal. Our initial submission was to ACPD, and the editor requested transfer to GMD, so we also defer to the editor's judgment in this case.

Lines 9-10: Isn't this point (coupling of microphysics and convection schemes) just a hypothesis provided at the end of the paper (Line 352-360)? If so, it doesn't belong in the abstract as a statement of certainty. I don't see any formal evidence presented to support this conjecture.

Yes, you're correct. We've replaced this text in the abstract with "*the broadening of the vertical velocity distribution with explicit convection.*" Figure 13 and Equations 4-13 do present concrete evidence for this point.

Thank you very much for highlighting this point, as it encouraged us to dig more into the coupling of microphysics and convection schemes, which we had misrepresented in Sec. 3.4.2. In fact, not even the condensate mass mixing ratios from the grid-scale microphysics are seen in the convective buoyancy formulation. We have corrected this text and now emphasize the separation of convective and grid-scale microphysics schemes in the final paragraph of Sec. 3.4.2.

Lines 23-25: Lu et al. (2007) do not discuss cloud-radiative impacts, and models do not agree on whether the presence of cloud radiative effects drive a poleward circulation shift (see discussion in Voigt et al. 2020 review). For example, Li et al. (2015) do not find a poleward expansion of the circulation due to the presence of cloud radiative effects, and they actually show that cloud-radiative effects decrease the static stability in the tropics.

Thank you for pointing out that this reference was not directly applicable. We remove this sentence and instead cite Li et al. 2015 and Voigt et al. 2020 for the idea that upper-tropospheric radiative heating in the tropics versus cooling in the midlatitudes promotes baroclinicity and static stability.

Line 32: The intensification of ENSO due to cloud radiative effects is again a model dependent result. Middlemas et al. (2019) found a differing effect on ENSO.
Thank you for noting that the language in this paragraph is too definitive. We add a reference to Middlemas et al. 2019 and clarify that "anomalies in cloud-radiative effects can intensify *or mute* the amplitude of ENSO *depending on model framework*."

Lines 136-137: More detail probably needs to be provided here to explain this conclusion, as the numbers in Table S1 do in fact look quite sensitive to the particular thresholds used.
Yes, there is a subtlety here. After describing the three sets of percentile thresholds (60-60-25, 62-67-30, and 65-70-35), we clarify that "*the cloud fractions associated with these percentile thresholds change by up to an order of magnitude; cloud fraction is generally larger than these threshold values when a cloud forms, so that the occurrence probability of cloud classes is mostly insensitive to which thresholds are used (Fig. S1).*"

Lines 146-155: It also seems important to note/discuss here that the altitude of the lower and upper tropospheric cooling peaks differs fairly significantly by model.
We have added a sentence to this paragraph to say that "*The altitudes of cloud-radiative cooling maxima also vary by about 80 hPa between the models in both the lower and upper troposphere.*"

Lines 169-170: Also convective heating rates appear to be important in this layer.
Yes, after the components mentioned (clear-sky LW cooling, dynamic heating, clear-sky SW heating, microphysical heating, and cloudy LW cooling), the convective heating would be next most important. We add a final sentence that "*the three smallest components of the budget are convective heating, shortwave cloud-radiative heating, and turbulent heating at these altitudes.*"

Lines 172-173: Also, the cooling peak appears to be slightly higher in altitude in the one-moment scheme.
Indeed. The goal with Figure 4 is primarily to indicate that LW cloud-radiative heating contributes non-negligibly to the heating budget both for the one- and two-moment schemes, so for simplicity's sake, we do not note this point in the text.

Line 180, typo: Change "Then" to "The"
Thank you, done.

Lines 184-185: Also, a large heating peak develops at lower altitudes, which is not present in the simulations with the deep convective parameterization.
Yes, this is worthy of mention, thank you. We state that "*the explicit representation of convection also produces prominent heating below 9 km, not present in the other two-moment simulations.*"

Lines 221-227: Good to double check the percentage values quoted in this paragraph. They appear to match what is shown in Fig. S3, not Fig. 7.

Many thanks for catching this. We showed the occurrence boxplots from the one-moment microphysics (Fig. S3) in an earlier draft and had not updated the values to reflect the two-moment boxplots.

Lines 230-232: Can you provide a physical explanation for why the isolated high clouds warm and the deeper clouds cool?

Yes, certainly. We have added the following: "*Isolated high clouds absorb more outgoing longwave radiation (OLR) than clear sky, whereas deep clouds absorb this OLR in the liquid cloud at lower altitudes and reemit it at colder temperatures from their cloud tops.*"

Line 243 (and hereafter): The term "higher grid spacing" could be confusing and could imply coarser resolution to some readers. I would either say "higher resolution" or "finer grid spacing".

Yes, this is a good point. We have changed instances of *higher grid spacing* to *finer grid spacing*.

Line 269: It doesn't look like a factor of four. At best, it looks like a factor of two.

Thank you for catching this. The max range of ice crystal number concentrations (2 $L^{-1}$ up to 8 $L^{-1}$) was mixed up with the max range of snow mass mixing ratio.

Line 271: The relative increase actually appears stronger in the thin cloud layers.

You are correct that the relative increase of $q_s$ is larger for the *High* and especially *High-Low* classes than either the *High-x-Middle* or *High-x-Middle-x-Low* classes. However, omission of $q_s$ from the CRH calculations is more influential for the deeper clouds with larger-magnitude $q_s$, since longwave absorption would be in proportion to absolute condensate mass. We clarify that the monotonic increase in $q_s$ has "*largest-magnitude changes from deep clouds.*"

Line 317, 325: This citation structure is confusing. Initially, I was looking for Fig. 10a and Table 2 in this paper. Please clarify that this figure and table are in the Sullivan et al. (2022) paper, and not this paper.

BibTeX is persnickety with citation formats like this. We have done the following: *(e.g. Sullivan et al., 2022, **their** Figure 10a)* and *(e.g. Sullivan et al., 2022, **their** Table 2).*

Line 328: I think you need to elaborate more on why you choose "supersaturation generated by vertical velocity" as one of your cloud controlling factors. The other two are obvious from the above equations, but this one is less obvious.

Thank you for point out this need for clarification. We note that "*T and $q_v$ appear explicitly in Eqs. 9-13, while the influence of w is felt indirectly by setting saturation with respect to ice ($RH_{ice}$ or $S_{ice}$ in Eqs. 11 and 13). The strength of w relative to $v_{Ts}$ also determines whether ice crystals sediment.*"

Line 384, typo: boundary

Thank you, done.

Figure 1 caption: North Africa, as well

Yes, thank you for catching this oversight.

Code and data availability: available is misspelled.
Thank you, corrected.

**References:**

Li, Y., Thompson, D. W. J., & Bony, S. (2015). The influence of atmospheric cloud radiative effects on the large-scale atmospheric circulation. *J. Clim.*, **8**, pp. 7263–7278. https://doi.org/10.1175/JCLI-D-14-00825.1

Middlemas, E. A., Clement, A. C., Medeiros, B., & Kirtman, B. (2019). Cloud radiative feedbacks and El Niño–southern oscillation. *J. Clim.*, **32**(15), pp. 4661–4680. https://doi.org/10.1175/JCLI-D-18-0842.1

---

## Author Comment (AC2)

**Reviewer 2**

Sullivan et al. (2023) analyze cloud radiative heating (CRH) in the upper troposphere over the North Atlantic Ocean as simulated by the ICON model. They provide a comprehensive analysis of dependencies on model resolution and ice-cloud microphysics parameterizations. They also attribute model differences to different classes of clouds with different vertical structure and to different cloud-controlling environmental factors, which I found useful for understanding the model differences. The paper is well written and logically organized, and the figures are clear. I have one suggestion for generalizing the results and several minor comments. I recommend that the manuscript be accepted if these minor issues are addressed.

We thank the reviewer for their time and effort in evaluating our work and for their feedback.

**General Comments**:

The authors comprehensively discuss CRH in the upper troposphere, but they barely mention CRH in the lower troposphere. However, previous studies have shown that the CRH climatology and relationship of CRH to prominent modes of natural variability both have a peak magnitude in the lower troposphere (Haynes et al., 2013; Papavasileiou et al. 2020; Wall et al., 2022). I realize that the authors want to focus on ice microphysics, for which it makes sense to analyze upper tropospheric CRH. However, I think neglecting lower-tropospheric CRH from the discussion gives the (unintentionally) misleading impression that it is less important for the overall CRH throughout the troposphere. Could the authors add a section that analyzes CRH for the low-level cloud classes and some discussion that compares this analysis with the CRH for the upper-level cloud classes? I think this would generalize the findings and add value to the paper.

Thank you for this feedback. We recognize the importance of the low-level CRH; indeed, in the North Atlantic climatological profiles shown in Figure 3a, the overall maximum in cloud-radiative cooling occurs below 850 hPa. However, our motivation is large-scale circulation, and we provide three reasons throughout to focus on the upper tropospheric values:

1. (Sec. 3.1) We do not perform global warming simulations here, but +4-K simulations from Voigt and Shaw 2015 and Voigt et al. 2019 indicate that the largest CRH differences with warming are localized in the upper troposphere. This increased warming enhances the meridional temperature gradient and expands the Hadley cell poleward.
   o Additionally, these same studies show that upper tropospheric CRH is more influential on circulation than boundary layer CRH when sea surface temperatures are prescribed as they are here.
2. (Sec. 3.1) Radiative cooling from water vapor constrains the top of the troposphere, not only in the tropics but also in the midlatitudes (Thompson et al. 2017). Given this constraint of clear-sky cooling on cloud-top temperatures and approximate cloud fractions, if we constrain upper tropospheric CRH in the current climate, we can also constrain it under warming. The same is not true for boundary-layer CRH.
3. (Introduction) Variability in the upper tropospheric CRH (between different models and between models and observations) is especially large [Voigt et al. 2019, Cesana et al. 2019]. In the tropics, the upper tropospheric variability is much larger than the boundary layer variability. This is less true in the midlatitudes.

However, we do not want to give the impression that low-level CRH should be neglected. To Sec. 3.1, we add "*Radiative cooling from extratropical low-level clouds has non-negligible effects on circulation, for example enhancing baroclinicity* [*Li et al. 2015*]."

**Specific Comments**:
(1)      Fig 12: It would help to plot all of the panels on the bottom row with the same range of vertical-velocity values.

Thank you for this suggestion, but we find that putting all the vertical velocity profiles on the same x-axis makes it somewhat more difficult to see the resolution dependence (panel 1 below). We do keep the x-axis limits uniform between the deeper (High-x-Middle *and* High-x-Middle-x-Low) and shallower (High *and* High-Low) cloud classes now, however (panel 2 below). We also note the different axis bounds in the caption now.

**panel 0 – original version**

[Figure]

**panel 1 – all xlim = [-0.75, 4]**

**panel 2 – consistent xlim for deeper versus shallower classes**

(2)      Fig 13: Has the vertical velocity data been horizontally averaged to a common scale prior to computing these histograms? If not, then is such a comparison meaningful? I would expect finer horizontal resolution to have larger variance of resolved vertical velocity but perhaps a different treatment of unresolved subgrid-scale vertical velocity by the convective parameterization scheme. It might help to discuss this around line 350.

No, the vertical velocities shown in Figure 13 have not been averaged / interpolated to the same grid across the simulations. Because we want to explain the differing $q_i$ profiles across grid spacings (among other factors), we also want to see the grid spacing dependence of these vertical velocities. Averaging or interpolating them to a uniform grid would be counterproductive. To the caption of Figure 13 we add "*Because we seek to explain the grid spacing dependence of $q_i$, these velocities are not averaged or interpolated to a uniform grid.*"

Then, as you say, finer horizontal resolution produces larger variance in the vertical velocity. The 5-km resolution simulations have the broadest spread in vertical velocities of the simulations with convective parameterization. We change wording in the description of Figure 13 to read "*the variance of these vertical velocity distributions becomes larger for finer grid spacing.*" Finally, to the discussion of Figure 12, we add a "*note that the ICON model uses no representation of subgrid scale variability in vertical velocities.*"

(3)      Line 376 states "Strong microphysical and convective sensitivity and weaker grid spacing sensitivity in the CRH profiles do not appear in distributions of cloud class occurrence and appear only weakly in cloud fraction profiles." Is this result a consequence of the fact that midlatitude synoptic weather systems are mostly resolved by all resolutions in the study? I'm just wondering if this result is specific to the midlatitudes or if it might generalize to the tropics. It would help to clarify this around line 376.

Yes, this is an interesting point. In looking through some previous studies of grid spacing sensitivity of extratropical cyclones, the general consensus seems to be that our coarsest resolution (80 km) is indeed sufficient to represent the macrostructure of the cyclone, i.e. its fronts and major airflows (e.g., Trzeciak et al. 2016, Flack et al. 2021, Priestly and Catto 2022). However, there is also substantial evidence that higher grid spacings produce stronger ascent rates, diabatic heating, and hence cyclone intensity (e.g., Willison et al. 2013, Flack et al. 2021, Choudhary and Voigt 2022). Whether increased intensity should generally imply increased cloud fraction or occurrence is another question.

Precisely because we trace the CRH dependencies back to differences in microphysical formulations, we do believe that these results are likely model-dependent. So we do not want to claim generalizability outside the midlatitudes, or even outside the ICON framework. To the paragraph in the Conclusions about robustness of findings, we add "*While our analysis method could be generalized to other regions or modeling frameworks, the role of $q_i$ and specific microphysical processes or parameters in CRH sensitivity will not necessarily generalize.*"

**References:**

(1) Haynes, J. M., Vonder Haar, T. H., L'Ecuyer, T., & Henderson, D. (2013). Radiative heating characteristics of Earth's cloudy atmosphere from vertically resolved active sensors. *Geophysical Research Letters*, *40*(3), 624–630. https://doi.org/10.1002/grl.50145

(2) Papavasileiou, G., Voigt, A., & Knippertz, P. (2020). The role of observed cloud-radiative anomalies for the dynamics of the North Atlantic Oscillation on synoptic time-scales. *Quarterly Journal of the Royal Meteorological Society*, *146*(729), 1822–1841. https://doi.org/10.1002/qj.3768

(3) Wall, C. J., Lutsko, N. J., & Vishny, D. N. (2022). Revisiting cloud radiative heating and the Southern Annular Mode. *Geophysical Research Letters*, 49, e2022GL100463. https://doi.org/10.1029/2022GL100463

**References:**

A. Voigt, N. Albern, and G. Papavasileiou (2019). The atmospheric pathway of the cloud-radiative impact on the circulation response to global warming: Important and uncertain. *J. Clim.* **32** (10) pp. 3051-3067.

A. Voigt and T. Shaw (2015). Circulation response to warming shaped by radiative changes of clouds and water vapour. *Nat. Geosci.* **8** pp. 102-106.

G. Cesana, D. E. Waliser, D. Henderson, T. S. L'Ecuyer, X. Jiang, and J. -L. F. Li (2019). The vertical structure of radiative heating rates: A multimodel evaluation using A-Train satellite observations. *J. Clim.* **32** pp. 1573—1590.

Y. Li, D. W. J. Thompson, and S. Bony (2015). The influence of atmospheric cloud radiative effects on the large-scale circulation. *J. Clim.* **28** (18) pp. 7263—7278.

T. M. Trzeciak, P. Knippert, J. S. R. Pirret, and K. D. Williams (2016). Can we trust climate models to realistically represent severe European windstorms? *Clim. Dyn.* **46** pp. 3431-3451.

D. L. A. Flack, G. Riviere, I. Musat, R. Roehrig, S. Bony, J. Delanoe, Q. Cazenave, and J. Pelon (2021). Representation by two climate models of the dynamical and diabatic processes involved in the development of an explosively deepening cyclone during NAWDEX. *Weather Clim. Dyn.* **2** pp. 233-253.

M. D. K. Priestly and J. L. Catto (2022). Improved representation of extratropical cyclone structure in HighResMIP models. *Geophys. Res. Lett.* **49** (5) e2021GL096708.

J. Willison, W.A. Robinson, and G. M. Lackmann (2013). The importance of resolving mesoscale latent heating in the North Atlantic storm track. *J. Atm. Sci.* **70** (7) pp. 2234—2250.

A. Choudhary and A. Voigt (2022). Impact of grid spacing, convective parameterization and cloud microphysics in ICON simulations of a warm conveyor belt. *Weather Clim. Dyn.* **3** (4) pp. 1199-1214.

---

## Author Comment (AC3)

**Editor Responses**

Dear authors,

In my role as Executive editor of GMD, I would like to bring to your attention our Editorial version 1.2: https://www.geosci-model-dev.net/12/2215/2019/

This highlights some requirements of papers published in GMD, which is also available on the GMD website in the 'Manuscript Types' section:

http://www.geoscientific-model-development.net/submission/manuscript_types.html

Thank you for bringing these guidelines to our attention.

In particular, please note that for your paper, the following requirements have not been fully met in the Discussions paper:

- "The main paper must give the model name and version number (or other unique identifier) in the title."
- "If the model development relates to a single model, then the model name and the version number must be included in the title of the paper. If the main intention of an article is to make a general (i.e. model independent) statement about the usefulness of a new development, but the usefulness is shown with the help of one specific model, the model name and version number must be state in the title. The title could have a form such as, "Title outlining amazing generic advance: a case study with Model XXX (version Y)"."
- "Code must be published on a persistent public archive with a unique identifier for the exact model version described in the paper or uploaded to the supplement, unless this is impossible for reasons beyond the control of authors. All papers must include a section, at the end of the paper, entitled "Code availability". Here, either instructions for obtaining the code, or the reasons why the code is not available should be clearly stated. It is preferred for the code to be uploaded as a supplement or to be made available at a data repository with an associated DOI (digital object identifier) for the exact model version described in the paper. Alternatively, for established models, there may be an existing means of accessing the code through a particular system. In this case, there must exist a means of permanently accessing the precise model version described in the paper. In some cases, authors may prefer to put models on their own website, or to act as a point of contact for obtaining the code. Given the impermanence of websites and email addresses, this is not encouraged, and authors should consider improving the availability with a more permanent arrangement. Making code available through personal websites or via email contact to the authors is not sufficient. After the paper is accepted the model archive should be updated to include a link to the GMD paper."

As your study was performed with ICON please add to the title this additional information (including a version number of ICON); e.g. "How Does Cloud-Radiative Heating over the North Atlantic Change with Grid Spacing, Convective Parameterization, and Microphysics Scheme? A case study with ICON vX.Y"

Additionally, state explicitly in you code availability section that ICON is license bound and thus can only be accessed via DWD (or MPI-M?)

We have added the ICON version number to the manuscript title and now note in the Acknowledgments that ICON is license bound.

Furthermore, if you did not use an official version but an intermediate version of the ICON model take care that this version is permanently archived (i.e with a DOI, e.g. at zenodo. Zenodo does not automatically mean, that the archive is open to the world. I understand that this is not possible, as long as ICON is not open source).

ICON version 2.1.00 was used for these simulations and is available by contacting DWD.

Last but not least, please put the scripts for reproducing the figures also on a permanent archive (github unfortunately is not permanent).

We have generated a GitHub release that is now archived in a Zenodo repository: https://zenodo.org/account/settings/github/repository/sylviasullivan/nawdex-hackathon

The URL in the acknowledgements has been corrected.

Yours,
Astrid Kerkweg